Clinical outcomes and risk factors of secondary extraintestinal manifestation in ulcerative colitis: results of a multicenter and long-term follow-up retrospective study

Xu Weimin 1
Ou Weijun 1
Guo Yuegui 1
Gu Yubei 2 gyb11809@rjh.com.cn
Cui Long 1
Zhong Jie 2
Du Peng 1 dupeng@xinhuamed.com.cn
1 Department of Colorectal Surgery, Xin-Hua Hospital, Shanghai Jiaotong University School of Medicine , Shanghai , China
2 Department of Gastroenterology, Rui Jin Hospital, affiliate to Shanghai Jiao Tong University, School of Medicine , Shanghai , China
Zhong Bao-Liang
Electronic publication date: 2019 Jun 21
Publication date: 2019
Volume: 7
Electronic Location ID: e7194
Received 2019 Feb 1; Accepted 2019 May 25
Copyright: © 2019 Xu et al.
Copyright year: 2019
Copyright holder: Xu et al.
License: This is an open access article distributed under the terms of the Creative Commons Attribution License, which permits unrestricted use, distribution, reproduction and adaptation in any medium and for any purpose provided that it is properly attributed. For attribution, the original author(s), title, publication source (PeerJ) and either DOI or URL of the article must be cited.
License URL: https://creativecommons.org/licenses/by/4.0/

Keywords: Ulcerative colitis, Extraintestinal manifestations, Clinical outcomes, Risk factors

Funding: National Natural Science Foundation of China 81873547 and 81570474 This work was supported by the National Natural Science Foundation of China (No. 81873547 and 81570474). The funders had no role in study design, data collection and analysis, decision to publish, or preparation of the manuscript.

==============================
Background

Extraintestinal manifestations (EIM) are common in ulcerative colitis (UC). In Shanghai, China, data on the incidence rate and risk factors of EIM in UC patients remain scarce.

Methods

The study population consisted of UC patients who were identified from a prospectively maintained, institutional review board-approved database at our institutes from June 1986 to December 2018. The demographic and clinical characteristics of the study participants were analyzed. The study included secondary EIM in UC patients and follow-up, while primary EIM was excluded. The diagnosis of EIM was based on clinical, radiological, endoscopic, and immunologic examination and histological findings.

Results

In total, 271 eligible patients were included in the current study, with a median follow-up time of 13.0 years (interquartile range, 9.0–17.0), and including 31 cases (11.4%) that developed EIM. EIM was associated with clinical outcomes in UC patients and the following factors were identified as contributing factors for the development of EIM: a disease duration of >5 years (odds ratio (OR), 3.721; 95% confidence interval (CI) [1.209–11.456]), age at diagnosis >40 years (OR, 2.924, 95% CI [1.165–7.340]), refractory clinical symptoms (OR, 4.119; 95% CI [1.758–9.650]), and moderate or severe anemia (OR, 2.592; 95% CI [1.047–6.413]).

Conclusion

In this study, approximately 11.4% UC patients go on to develop at least one EIM. Clinicians should prioritize early control of the disease and treatment of anemia in UC in order to prevent the development of EIM and improve disease prognosis.

Introduction

As a major form of inflammatory bowel disease (IBD), ulcerative colitis (UC) is a multifactorial polygenic disease with probable genetic heterogeneity, which is characterized by a chronic course of recurrent relapse and remission (Fiocchi, 1998; Podolsky, 2002; SC, 2017). In clinical practice, UC patients often present with complex and diverse gastrointestinal symptoms combined with relatively rare extraintestinal manifestations (EIM). The clinical presentation of EIM is extremely heterogeneous and can affect almost all of the organs in the body; EIM is especially involved in blood, joints, skin, eyes, biliary tracts, the vascular system, and organs such as the liver, kidneys, and lungs. Some diseases, such as oral aphthous ulcers and erythema nodosum, parallel the activity of the bowel disease, but for a number of these conditions, such as ankylosing spondylitis, they follow an independent course from the disease activity of UC (Rothfuss, Stange & Herrlinger, 2006; Trikudanathan, Venkatesh & Navaneethan, 2012). All these clinical symptoms refer to the secondary EIM, while primary EIM refers to extraintestinal symptoms that occur before the onset of UC and are not related to UC.

Extraintestinal manifestations affects the outside gastrointestinal tract and some forms are associated with UC as a result of autoimmune diseases such as rheumatoid arthritis and vasculitis; these diseases can sometimes mimic each other, making differential diagnosis difficult for clinicians (Colia, Corrado & Cantatore, 2016). Moreover, multiple EIM may occur concomitantly, and the presence of a single EIM confers a higher likelihood of developing additional EIM (Vavricka et al., 2011), which seriously compromises the patients’ quality of life and aggravates the disease. Therefore, early diagnosis and prompt intervention are imperative for EIM in clinical practice. However, few studies have specifically reported the clinical outcomes, especially the risk factors, of EIM in the Chinese population.

In the present study, we mainly research the incidence rate of EIM in Shanghai city and the associations between EIM and clinical outcomes. Additionally, we also aim to determine the risk factors contributing to the development of EIM, which could aid in earlier identification, and further prevent the development of EIM.

Methods

Study population

All consecutive UC patients at the Department of Colorectal Surgery, Xin-Hua Hospital, Shanghai Jiao Tong University School of Medicine (Shanghai, China), and the Department of Gastroenterology, Rui-Jin Hospital, Shanghai Jiao Tong University School of Medicine (Shanghai, China) from June 1986 to December 2018 were enrolled in the study. Patients were identified and their clinical data was obtained from a prospectively maintained, institutional review board-approved database (Chinese Database System for IBD, CHASE-IBD). The Ethics Committee of Xin-hua Hospital approved this study (approval no. XHEC-D-2018-089).

Inclusion and exclusion criteria

Inclusion criteria were as follows: (1) positive diagnosis of UC; (2) age at diagnosis ≥18 years old; and (3) received a regular follow-up at our department. Exclusion criteria were as follows: (1) patients who were diagnosed with familial adenomatous polyposis or indeterminate colitis; (2) patients with poor compliance; and (3) patients with underlying disease, impaired general health, and/or lost to follow-up.

In this study, poor patient compliance refers to patients who refuse to receive regular follow-up and give feedback to the doctor. Underlying disease was mainly defined as the infectious factors caused of intestinal inflammation as well as colorectal cancer (CRC) or high-grade dysplasia. The impaired general health refers to serious and chronic diseases such as hypertension, diabetes, hyperthyroidism and heart failure, which lead to poor general conditions.

Data collection and clinical evaluation

The patient’s medical history, examination data, surgical information, and clinical outcomes were retrospectively collected from the hospital medical records, outpatient examination, and long-term regular follow-up. The collected data were as follows: sex, age at diagnosis, disease duration, relapse, EIM, gastrointestinal symptoms, weight loss, primary sclerosing cholangitis (PSC), disease extent according to the Montreal classification system (Silverberg et al., 2005), surgical history, familial history of IBD or CRC, complications, development of CRC, use of steroids and immunomodulators, and hemoglobin (Hb) and Albumin (Alb) measurements. Each patient will undergo routine and systematic imaging examinations, colonoscopy or biopsy, immunological examinations, etc. All these baseline characteristics were presented based on the most recent follow-up. Relevant data of EIM was collected during treatment. In addition, the use of mesalamine, biologics, steroids, and immunomodulators refer to current treatment or the previous treatment that was used until the latest follow-up (December 2018). UC was strictly diagnosed according to the established internationally accepted criteria proposed by Lennard-Jones (1989). The criteria excluded infectious factors caused of intestinal inflammation. The diagnosis of UC and EIM were based on clinical, radiological, endoscopic, and immunologic examination, and histologic findings. In this study, primary EIM refers to extraintestinal symptoms that occur before the onset of UC and are not related to UC. While secondary EIM refers to extraintestinal symptoms that occur in the course of UC that affect various organs in the body. It includes various clinical manifestations such as rash, oral aphthous ulcers, arthritis, glomerulonephritis, PSC, vasculitis, ankylosing spondylitis, and autoimmune hepatitis. Only when these extraintestinal symptoms appear in the course of UC, and after detailed radiological, immunologic and histologic examination, can be diagnosed as a secondary EIM. The rash in the text refers to the long-term persistent skin damage that occurs in the course of UC, while the transient rash that occurs after drug treatment is not taken into account. We defined refractory clinical symptoms as more than 5 stools per day accompanied by diarrhea and mucous, and bloody stools after regular medical treatment, including 5-aminosalicylic acids (5-ASA), corticosteroid, immunomodulators, and biologics. In this study, patients with “weight loss” refers to patients who lost more than five kg at the latest follow-up. Family history of other cancers refers to first, second or third degree relative of the patient had or is currently living with cancer except CRC. Mild anemia was defined as Hb value was less than 120 g/L but more than 90 g/L in male and less than 110 g/L but more than 90 g/L in female. Moderate anemia refers to Hb value between 60 and 90 g/L, and severe anemia was defined as Hb value <60 g/L both in male and female, respectively. Therefore, in this study, we modified the definition that Hb <90 g/L refers to moderate or severe anemia. Patients with Hb value <90 g/L has at least a moderate anemia even more serious anemia. Remission in this study refer to the relief of clinical symptoms including less than three times of stools per day, the disappearance of diarrhea, fever, abdominal bloating and distension, and stool with mucus, pus and blood. To evaluate the associations between EIM and clinical outcomes, we divided the patients into two groups with or without EIM. The complications refer to various clinical manifestations as well as the development of CRC. All complications were diagnosed based on clinical manifestations, laboratory results, and endoscopic and imaging findings. Intestinal obstruction, colon perforation and gastrointestinal bleeding were defined as the serious complications.

Statistical analysis

SPSS version 19.0 software (IBM Corp., Armonk, NY, USA) were used for the statistical analyses. Values and percentages, mean and standard deviations, or the median and interquartile range (IQR) were used to describe the different varieties of data, respectively. Chi-squared or Fisher’s exact test were used for categorical variables. Moreover, we choose the Wilcoxon’s rank-sum test as the statistical method for ranked variables. Multivariate logistic regression was performed in the variables with p-value <0.05 in the analysis of the risk factors of EIM in UC. Confidence intervals (CI) were set at 95%. All statistical tests were two-sided, with a p-value of <0.05 considered statistically significant.

Results

Demographics and clinical characteristics

A total of 291 UC patients were enrolled at our institute, 20 patients were lost to follow-up and excluded; 271 eligible patients were ultimately included for the current research. A schematic flow diagram of the present study is shown in Fig. 1 and the patient demographics and clinical and laboratory characteristics are shown in Table 1. In the entire cohort, the median follow-up time was 13.0 years (IQR: 9.0–17.0 years) from June 1986 to December 2018. Among the 271 patients, 132 were males and patients had a median age at diagnosis of 42.0 years (IQR: 29.0–53.3 years) and a median disease duration of 7.0 years (IQR: 4.0–10.0 years). In the entire cohort, 31 patients (11.4%) were found to develop at least one EIM, of which eight patients (3.0%) developed multiple EIM. In addition, six patients (2.2%) had proctitis (E1), 140 (51.7%) had left-sided colitis (E2), and 125 (46.1%) had pancolitis (E3). In addition, in the total of 31 EIM patients, 14 were men. Median age at diagnosis and disease duration in EIM patients was 47.0 years (IQR: 44.0–55.0 years) and 14,0 years (IQR:6.0–17.0 years), respectively (Table 1).

Figure 1 A schematic flow diagram of the present study.

Table 1 Patient characteristics at the most recent follow-up visit.

Variables	Non-EIM group (n = 240)	EIM group (n = 31)	All cases (n = 271)	
Sex (male/female)	118/122	14/17	132/139	
Age at diagnosis [year, median (IQR)]	41.0 (28.0–53.0)	47.0 (40.0–55.0)	42.0 (29.0–53.3)	
Disease duration [year, median (IQR)]	6.0 (4.0–10.0)	14.0 (6.0–17.0)	7.0 (4.0–10.0)	
Follow-up time [year, median (IQR)]	13.0 (9.0–17.0)	9.0 (6.0–13.0)	13.0 (9.0–17.0)	
Relapse, n (%)	
 First occurrence	57 (23.8)	6 (19.4)	63 (23.2)	
 First recurrence	47 (19.6)	2 (6.5)	49 (18.1)	
 Multiple recurrence	136 (56.7)	23 (74.2)	159 (58.7)	
Stool with mucous, n (%)	112 (46.7)	21 (67.7)	133 (49.1)	
Bloody stool, n (%)	207 (86.3)	25 (80.6)	232 (85.6)	
Abdominal bloating and distension, n (%)	48 (20.0)	9 (29.0)	57 (21.0)	
Diarrhea	159 (66.3)	28 (90.3)	187 (69.0)	
Stool frequency, n (%)	
 <4 times	204 (85.0)	18 (58.1)	222 (81.9)	
 ≧4 time	36 (15.0)	13 (41.9)	49 (18.1)	
Weight loss, n (%)	86 (35.8)	16 (51.6)	102 (37.6)	
Extraintestinal manifestations (EIMs), n (%)		31 (11.4)	31 (11.4)	
 Single EIMs		23 (8.4)	23 (8.4)	
 Multiple EIMs		8 (3.0)	8 (3.0)	
History of surgery, n (%)	30 (12.5)	7 (22.6)	37 (13.7)	
Family history, n (%)	15 (6.3)	3 (9.7)	18 (6.6)	
 Family history of CRC	3 (1.3)	1 (3.2)	4 (1.5)	
 Family history of IBD	2 (0.8)	1 (3.2)	3 (1.1)	
 Family history of other caner	12 (5.0)	1 (3.2)	13 (4.8)	
Extent of UC, n (%)	
 E1	6 (2.5)	0 (0.0)	6 (2.2)	
 E2	125 (52.1)	16 (51.6)	141 (52.0)	
 E3	109 (45.4)	15 (46.9)	124 (45.8)	
Probiotics, n (%)	84 (35.0)	11 (35.5)	95 (35.1)	
Steroids, n (%)	148 (61.7)	21 (67.7)	169 (62.4)	
Immunomodulators, n (%)	32 (13.3)	2 (6.5)	34 (12.5)	
Hb (g/L, mean ± SD)	114.5 ± 25.1	109.7 ± 27.1	113.9 ± 25.3	
ALT (g/L, mean ± SD)	28.6 ± 23.1	24.7 ± 11.6	28.2 ± 22.1	
Note:

CRC, colorectal cancer; UC, ulcerative colitis; IQR, interquartile range; Hb, hemoglobin; IBD, inflammatory bowel disease; Alb, Albumin.

Analysis of EIM in the present study

Table 2 showed the detailed EIM in UC patients. In the 23 patients with single EIM, nine patients developed arthritis, six had oral aphthous ulcers, four had rash, two experienced glomerulonephritis, one had PSC, and one had ankylosing spondylitis. For eight patients with multiple EIM, three patients had arthritis and oral aphthous ulcers, one had arthritis and rash, one had oral aphthous ulcers and rash, one had arthritis and iritis, one developed rash and vasculitis, and one experienced PSC and autoimmune hepatitis (Table 2). Therefore, this data indicated that the overall incidence rate of EIM of UC in Shanghai was 11.4%.

Table 2 Analysis of extraintestinal manifestations (EIM).

EIM	Non-EIM group (%)	EIM group (%)	N (%)	
Rash	0 (0.0)	7 (22.6)	7 (2.6)	
Oral aphthous ulcers	0 (0.0)	10 (32.3)	10 (3.7)	
Arthritis	0 (0.0)	14 (45.2)	14 (5.2)	
Primary sclerosing cholangitis (PSC)	0 (0.0)	2 (6.5)	2 (0.7%)	
Iritis	0 (0.0)	1 (3.2)	1 (0.4)	
Vasculitis	0 (0.0)	1 (3.2)	1 (0.4)	
Ankylosing spondylitis	0 (0.0)	1 (3.2)	1 (0.4)	
Autoimmune hepatitis	0 (0.0)	1 (3.2)	1 (0.4)	
Glomerulonephritis	0 (0.0)	2 (6.5)	2 (0.7)	

Analysis of complications in UC

In the entire cohort, 107 patients (39.5%) had complications during hospitalization or follow-up. The most common complication was inflammatory polyps, which occurred in 63 patients (23.2%). In addition, 28 (10.3%) experienced gastrointestinal bleeding, 21 (7.7%) demonstrated intestinal obstruction, and 6 (2.2%) had colon perforation. Intestinal obstruction, colon perforation and gastrointestinal bleeding were defined as the serious complications. Moreover, 11 patients (4.1%) went on display malignant transformation and were subsequently diagnosed with CRC. In EIM patients, eight patients (25.8%) developed colorectal stricture, 5 (16.1%) had intestinal obstruction, and 4 (12.9%) had CRC (Table 3).

Table 3 Analysis of complications in ulcerative colitis (UC) patients.

Complications	Non-EIM group (%)	EIM group (%)	N (%)	
Colorectal stricture	28 (11.7)	8 (25.8)	36 (13.3)	
Bleeding	24 (10.0)	4 (12.9)	28 (10.3)	
Inflammatory polyps	57 (23.8)	6 (19.4)	63 (23.2)	
Intestinal obstruction	16 (6.7)	5 (16.1)	21 (7.7)	
Colon perforation	5 (2.1)	1 (3.2)	6 (2.2)	
Toxic megacolon	2 (0.8)	0 (0.0)	2 (0.7)	
Abscess formation	3 (1.3)	0 (0.0)	3 (1.1)	
CRC	7 (2.9)	4 (12.9)	11 (4.1)	
Note:

CRC, colorectal cancer; EIM, extraintestinal manifestation.

Comparison of clinical outcomes between UC patients with and without EIM

Clinical outcomes including remission, serious complications (intestinal obstruction or perforation or bleeding), surgery, development of CRC, and death were analyzed in non-EIM group and EIM group. As shown in Table 4, remission was more common in the non-EIM group (n = 127, 52.9%) than in the EIM group (n = 8, 25.8%) (p = 0.004). Patients with EIM had a higher likelihood of developing serious complications than those in the non-EIM group (n = 10, 32.3% vs n = 40, 16.7%, p = 0.035). In addition, patients in the EIM group had a higher risk of developing CRC than those in the non-EIM group (n = 4, 12.9% vs n = 7, 2.9%, p = 0.030). There were no statistical differences between the two groups with regards to cure, disease activity, surgery, or death. Taken together, this result suggested that EIM was closely associated with clinical outcomes in UC.

Table 4 Comparison of clinical outcomes between ulcerative colitis patients with and without extraintestinal manifestations (EIMs).

Variables	Non-EIM group	EIM group	p-value	
Remission, n (%)			0.004a	
 No	113 (47.1)	23 (74.2)		
 Yes	127 (52.9)	8 (25.8)		
Serious complications, n (%)			0.035a	
 No	200 (83.3)	21 (67.7)		
 Yes	40 (16.7)	10 (32.3)		
Surgery, n (%)			0.160a	
 No	150 (62.5)	19 (61.3)		
 Yes	90 (37.5)	12 (38.7)		
CRC, n (%)			0.030b	
 No	233 (97.1)	27 (87.1)		
 Yes	7 (2.9)	4 (12.9)		
Death, n (%)			0.718b	
 No	232 (96.7)	29 (93.5)		
 Yes	8 (3.3)	2 (6.5)		
Notes:

CRC, colorectal cancer.

a Chi-squared.

b Fisher’s exact test. Serious complications: Intestinal obstruction, colon perforation and gastrointestinal bleeding.

Risk factors for the development of EIM

Based on the previous results, we determined which risk factors were associated with the development of EIM. The results of univariable analysis showed that the age at diagnosis (p = 0.011), disease duration (p = 0.013), refractory clinical symptoms (p < 0.001), and the presence of moderate or severe anemia (p = 0.024) were all significantly associated with the development of EIM. No significant differences were observed between the EIM group and the non-EIM group in terms of sex, relapse, weight loss, history of surgery, family history, extent of UC, use of steroids and immunomodulators, and the Alb level (Table 5). The results of multivariate logistic analysis demonstrated that a disease duration >5 years (odds ratio (OR), 3.721; 95% CI [1.209–11.456]; p = 0.022), an age at diagnosis >40 years (OR, 2.924, 95% CI [1.165–7.340]; p = 0.022) and refractory clinical symptoms (OR, 4.119; 95% CI [1.758–9.650]; p = 0.001) were associated with the development of EIM. In addition, moderate or severe anemia (OR, 2.592; 95% CI [1.047–6.413]; p = 0.039) was associated with the development of EIM (Table 6).

Table 5 Risk factors for extraintestinal manifestations (EIMs) in ulcerative colitis patients.

Variables	Non-EIM group	EIM group	p-value	
Sex, n (%)			0.675a	
 Male	118 (49.2)	14 (45.2)		
 Female	122 (50.8)	17 (54.8)		
Age at diagnosis, n (%)			0.011a	
 <40 years	112 (46.7)	7 (22.6)		
 ≧40 years	128 (53.3)	24 (77.4)		
Disease duration, n (%)			0.013a	
 <5 years	84 (35)	4 (12.9)		
 ≧5 years	156 (65)	27 (87.1)		
Relapse, n (%)			0.117b	
 First occurrence	57 (23.8)	6 (19.4)		
 First recurrence	47 (19.6)	2 (6.5)		
 Multiple recurrence	136 (56.7)	23 (74.2)		
Refractory clinical symptoms, n (%)			<0.001a	
 No	204 (85.0)	18 (58.1)		
 Yes	36 (15.0)	13 (41.9)		
Weight loss, n (%)			0.088a	
 No	154 (64.2)	15 (48.4)		
 Yes	86 (35.8)	16 (51.6)		
History of surgery, n (%)			0.208a	
 No	210 (87.5)	24 (77.4)		
 Yes	30 (12.5)	7 (22.6)		
Family history, n (%)			0.346a	
 No	225 (93.8)	28 (90.3)		
 Family history of CRC or IBD	5 (2.1)	2 (6.5)		
 Family history of other cancers	10 (4.2)	1 (3.2)		
Extent of UC, n (%)			0.690b	
 E1	6 (2.5)	0 (0.0)		
 E2	124 (51.7)	16 (51.6)		
 E3	110 (45.8)	15 (48.4)		
Steroids, n (%)			0.511a	
 No	92 (38.3)	10 (32.3)		
 Yes	148 (61.7)	21 (67.7)		
Immunomodulators, n (%)			0.423a	
 No	208 (86.7)	29 (93.5)		
 Yes	32 (13.3)	2 (6.5)		
Hb, n (%)			0.024a	
 ≧90 g/L	202 (84.2)	21 (67.7)		
 <90 g/L	38 (15.8)	10 (32.3)		
Alb, n (%)			0.202a	
 ≧35 g/L	90 (37.5)	8 (25.8)		
 <35 g/L	150 (62.5)	23 (74.2)		
Notes:

CRC, colorectal cancer; UC, ulcerative colitis; Hb, hemoglobin; IBD, inflammatory bowel disease; Alb, Albumin.

a Chi-squared or Fisher’s exact test.

b Wilcoxon’s rank-sum test.

Table 6 Multivariate logistic regression analysis of risk factors for extraintestinal manifestations (EIMs) in ulcerative colitis patients.

Variable	Univariatea	Multivariateb	
Odds ratio	95% CI	p-value	Odds ratio	95% CI	p-value	
Disease duration (≧5 years)	3.635	[1.231–10.735]	0.020	3.721	[1.209–11.456]	0.022	
Age at diagnosis (≧40 years)	3.000	[1.245–7.228]	0.014	2.924	[1.165–7.340]	0.022	
Refractory clinical symptoms	4.093	[1.875–9.077]	0.001	4.119	[1.758–9.650]	0.001	
Moderate anemia	2.531	[1.105–5.799]	0.028	2.592	[1.047–6.413]	0.039	
Relapse	1.418	[0.858–2.342]	0.173				
Notes:

CI, confidence interval.

a Univariate logistic regression analysis.

b Multivariate logistic regression analysis.

Discussion

It is a challenge for clinicians to accurately diagnose EIM due to the complexity and diversity of its clinical manifestations. Although EIM is relatively rare and more frequently ignored compared to the gastrointestinal symptoms of UC, it tends to aggravate the severity of disease leading to a poor prognosis. There are a limited number of studies that have researched the effect of EIM on clinical outcomes, especially the risk factors for the development of EIM in the Chinese UC population. To the best of our knowledge, this study is the largest multicenter study in Shanghai to evaluate the clinical outcomes and risk factors of EIM with a long-term follow-up from June 1986 to December 2018. The major findings of this study can be summarized as follows: (1) The overall incidence rate of EIM of UC in Shanghai was 11.4%. (2) Patients with EIM had worse clinical outcomes, which mainly presented as lower remission rates, increased incidence of serious complications, and more likely to develop CRC. (3) A disease duration >5 years, an age at diagnosis >40 years, refractory clinical symptoms, and moderate or severe anemia contributed to the development of EIM.

The epidemiology of EIM varies widely in different reports, with overall prevalence rates ranging from 6% to 40% (Bernstein et al., 2001; Lakatos et al., 2003; Vavricka et al., 2011; Zippi et al., 2014). Zippi et al. (2014) reported the highest prevalence of 40.6% in Italy and Bernstein et al. (2001) reported the lowest prevalence of 6.2% in Canada. Moreover, Switzerland, Hungary, and Portugal had prevalence rates of 38.1%, 21.3%, and 25.8% (Lakatos et al., 2003; Vavricka et al., 2011; Veloso, Carvalho & Magro, 1996), respectively. The over incidence of EIM in Shanghai is only slightly higher than the rate reported in the Canadian retrospective study. These variations may be attributed to specific characteristics of the study populations, differences in the study design and inclusion criteria, accuracy of diagnosis, and previous medical treatment. In the case of EIM, our results are similar to previous studies (Isene et al., 2015) in that arthritis and oral aphthous ulcers. Interestingly, a previous study in Wuhan, Central China reported a 5.7% prevalence of EIM in UC (Jiang et al., 2006), which indicated that different regions of the same country may also have different incidence rates of EIM. Taken together, the overall extent of EIM of UC in China is lower than that in western countries. This may be associated with the different ethnic and environmental factors and it may be beneficial in future studies to determine the genetic and environmental factors involved in the etiology and pathogenesis of EIM.

Despite the lower incidence of EIM in China compared to western countries, EIM can increase the risk of poor clinical outcomes, such as lower remission rates, an increase in serious complications, and even malignant transformation. Therefore, it is imperative for clinicians to uncover the risk factors for the development of EIM earlier and conduct appropriate disease surveillance in clinical practice.

A certain proportion of UC patients presented with serious gastrointestinal symptoms such as diarrhea, stools with blood and mucus, and fecal incontinence. After regular treatment, including the most commonly used 5-ASA (Schroeder, Tremaine & Ilstrup, 1987), corticosteroid, and immunomodulators, most patients experienced remission, although some patients were still associated with active disease. Although the successful application of anti-TNF-α treatment was a major breakthrough in the treatment of IBD, approximately one-third of patients do not respond to anti-TNF-α treatment, and many others eventually lose responsiveness or become intolerant to these agents (Melmed & Targan, 2010). Noteworthily, previous researches reported that EIM was a flare (Das, 1999; Su, Judge & Lichtenstein, 2002). As for the characteristic breakdown of intestinal homeostasis in UC, this is considered to arise from a complex interaction of immunological and environmental factors in a genetically predisposed individual. Therefore, we speculated that heterogeneity between individuals led to different presentations of clinical symptoms, which in turn, could aggravate disease activity toward the development of EIM.

The prevalence and severity of anemia are related to the activity of the bowel disorder and a recent meta-analysis reported that anemia occurred in 21% of UC patients in Europe (Vegh et al., 2016). In this study, a total of 48 patients (17.7%) experienced moderate anemia; this was slightly lower than the previously reported value of 21%. One reason for this could be because we defined “anemia” as “moderate or severe anemia” with a Hb level <90 g/L. As a retrospective study with 1,158 Crohn’s disease (CD) and 1,108 UC patients reported, 19.6% CD and 21.6% UC patients were diagnosed with iron deficiency anemia (IDA). In the CD group, the Crohn’s disease activity index (CDAI) was considerably higher in patients suffering from IDA, while UC patients with IDA showed a significantly higher rate of erythema nodosum, which is kind of disease activity-related EIM (Madanchi et al., 2018). The results of the current study indicate that anemic patients have a higher disease severity. Furthermore, a previous study demonstrated that anemia in IBD patients was significantly associated with increased ESR, CRP, and CDAI, and that the disease activity scores showed an inverse correlation with the Hb level (Bergamaschi et al., 2010). Taken together, UC patients with moderate anemia usually present with higher disease activity which in turn, aggravates the anemia; it is possible that this interaction is responsible for the development of EIM.

There are several limitations to our study. First, it is possible that selection bias was present since all included patients in the present study were diagnosed and managed by colorectal surgeons and gastroenterology clinicians in two different departments. Second, since this is a retrospective study, a loss of follow-up is inevitable. Third, the sample size is relatively small, so the conclusion we drew is not accurate enough. A larger sample of multiple-center studies should be performed in the near future. On the other hand, there are certain strengths to our study. This study is a retrospective study in Shanghai to evaluate the incidence rate of EIM in UC patients. We confirmed the negative effect of EIM on the clinical outcomes. It is more important that we first identified the risk factors associated with EIM in UC, which helps early intervention to prevent the occurrence of EIM.

Conclusion

The present study showed that the incidence rate of EIM of UC in Shanghai was 11.4%. Furthermore, we also discovered that EIM was associated with clinical outcomes in UC patients. In addition, a disease duration >5 years, an age at diagnosis >40 years and refractory clinical symptoms were also found to be contributing factors for the development of EIM, while moderate or severe anemia was associated with EIM. Therefore, appropriate and effective treatment to control active UC and treatment of anemia to improve general condition are imperative in order to prevent the development of EIM and further improve the long-term outcomes of UC patients.

Supplemental Information

Supplemental Information 1 Raw data.

The clinical and basic characteristics of the whole cohort (271 patients). All data was collected from electronic medical records and follow-up. All statistical analysis is performed on this data.

Click here for additional data file.

Additional Information and Declarations

Competing Interests

Author Contributions

Ethics

Data Availability

The authors declare that they have no competing interests.

Weimin Xu performed the experiments, analyzed the data, prepared figures and/or tables, authored or reviewed drafts of the paper, approved the final draft.

Weijun Ou performed the experiments, prepared figures and/or tables, approved the final draft.

Yuegui Guo analyzed the data, approved the final draft.

Yubei Gu conceived and designed the experiments, approved the final draft.

Long Cui contributed reagents/materials/analysis tools, approved the final draft.

Jie Zhong contributed reagents/materials/analysis tools, approved the final draft.

Peng Du conceived and designed the experiments, authored or reviewed drafts of the paper, approved the final draft.

The following information was supplied relating to ethical approvals (i.e., approving body and any reference numbers):

The Ethics Committee of Xin-hua Hospital approved this study (approval no. XHEC-D-2018-089).

The following information was supplied regarding data availability:

The raw measurements are available in the Supplemental Files.

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
