# Peer review of "Clinical outcomes and risk factors of secondary extraintestinal manifestation in ulcerative colitis: results of a multicenter and long-term follow-up retrospective study"

_PeerJ, doi:10.7717/peerj.7194_

## Round 0.1 · original submission · Major Revisions

Although the study has merits, it still has much room for improvement. Comments from our reviewers are sufficiently detailed, please kindly consider their comments and revise the paper accordingly.

Reviewer 1 ·

Basic reporting

The overall English language is acceptable. However, there are some inappropriately placed terms, e.g.: line 205 accompanied by should be (associated with), line 209:associated with disease activity, should be (a flare)

Literature references ,sufficient felid background/ context were provided. However, two of the main references they used for discussion were not open access articles and were difficult to obtain

Professional article structure, raw data shared. However, table 4 had a missing p value for the variable surgery, the flow chart could be simplified

Self contained with relevant results to hypothesis was described

Experimental design

It is an original article within the aims and scopes of the journal.

Research question was not similarly defined in abstract and introduction. However, it could be understood from the sequence. Additionally, lines 60,61 were little difficult to understand, They didn't assess the severity of EIM but they did check the clinical outcomes according to EIM or non EIM.

The knowledge gap was well stated and they designed their study to cover this gap

investigations performed to ethical standards

methods: were described but there are some issues:
(lines 87):They used the Montreal classification system then they described all types of symptoms manifested by the patients. It would be easier, probably more valid and statistically significant, if they calculated the S1,S2 and S3 and include it in the analyses

(lines 95,96 They defined weight loss as losing more than 5 kg over study period (mean 13 years of follow up)

(Line 99) is the should not refer to statistics in other paper, instead write it.

( line 139) should be moved to the method. We suddenly find in the results that they compared the groups with no prior explanation in the methods

(line 104) they didn't describe in statistics the univarite analysis used

I expected to find a definition for anemia grades in methods not in the line 220 in the discussion

Table 1 family history of other cancers were analysed for but it was not included in the methods

Validity of the findings

Repetition of results both in text(in the results and discussion) and tables except for table 5 and table 6 is noted. Even tables 1 and 5 include repeats, instead keep table 5 with the p values

Table 5 it seems that the statistical method used was chi square and student t test. I don't believe they are univariate analyses. It is better to state which statistic test used under each table.

conclusions are stated ,linked to original research question and supporting results.However, it includes a term in line 244(early appropriate and effective comprehensive) it sounds over structured

Additional comments

Thank you for submitting your research, I can see that you made an big effort to collect and analyze your data. I would also thank you for providing your raw data. However, I suggest you organize your paper a little bit:
1- clearly state your aim.
2-clearly describe your methods and get it more organized, don't under-write in statistics part. It is important that the readers understand how you did your research.
-I have a question in line 90 ,why did you choose to exclude primary EIM? in such case how many patients were they and why they were not described on the figure1? on table 1 are the single EIM (single secondary EIM)

3-When you report your results avoid unnecessary repletion: I would suggest cleaning the figure 1 ,keep tables 5 and 6 and state which type of statistical test you have used, if you wish to keep table 4 add the p value on surgery and don't repeat it in the text.
- It is better to mention the sites for single or multiple EIM.
Table 3 why did you include CD patient in the analysis, if he was a UC then moved into CD this should be excluded.
-Keep the( medical refractory disease) standardized allover the manuscript don't use the alternate or continuous disease activity in table 4 don't us
- In line 124, why did you consider PSC as an EIM?
line 134 did you mean colon or rectal stricture? How severe was the gastro-intestinal bleeding? was it endoscopically diagnosed or a clinically manifested bleeding?

4-limitation of the study: you did report some study limitations on the lines 232-235. You didn't not comment on missing data and how to handle it, was there any miss recording.

5- I suggest more thorough revision of English language

Reviewer 2 ·

Basic reporting

The paper is relatively easy to follow with clear and unambiguous English used throughout. References seem to be outdated with not so many most recent and relevant papers cited. The article is structered in professional way.

Experimental design

Original primary research is within Aims and Scope of the journal. The methodology has several drawbacks. Methods are described with insufficient details. Some important data are lacking (e.g. reporting of disease activity index, endoscopic disease activity index, C-reactive protein)

Validity of the findings

The study population consisted of UC patients who were identified from a prospectively maintained, institutional review board-approved database at Authors institutes from
June 1986 to December 2018 and 271 ulcerative colitis patients were included into the study. It is difficult to extrapolate these results to the whole population of Shanghai, eastern China as authors state in conclusion: “In Shanghai, eastern China, approximately 11.4% UC patients go on to develop at least one EIM”. It is not clear from the paper what was the baseline population of patients. Were these patients admitted to surgery word due to ongoing complications (e.g. severe bleeding ?). Where these patients admitted with suspicion of ulcerative colitis or signs of bleeding ? Where the study group outpatients ?

Additional comments

Other comments:


The demographic and clinical characteristics of the study participants were analyzed, but there is no information of disease activity index or endoscopic disease activity index. There is no information on C-reactive protein (CRP) or any other inflammatory indicators. The clinical remission (> stools per day plus other symptoms such as bloody diarrhea) was the only measure to evaluate whether patients were in remission or not. It seems that significant number of patients might have been qualified as in remission in fact without remission.

Mean age of diagnosis is relatively high 42.0 (29.0–53.3). Where are the younger patients ? Were they managed by other units ? Outpatient clinics ?

The study included secondary EIM in UC patients and follow-up, while primary EIM was excluded. Please explain what are primary and secondary EIM as it is not clear in the draft.
(For example: This study focused on secondary EIM in UC disease course and follow-up, while primary EIM was excluded) The diagnosis of EIM was based on clinical, radiological, endoscopic, and immunologic examination and histological findings. No information and data are provided as for these diagnostic tests used (how many or which patients had which test performed)

All complications were diagnosed based on clinical manifestations, laboratory results, and endoscopic and imaging findings – for example bleeding is reported by authors as complication (10.3% experienced gastrointestinal bleeding – upper or lower GI ?), but bleeding is also reported as necessary clinical symptom of no remission. It is impossible to find out from the study whether bleeding was a complication or natural course of disease in non-remission. There is lack of information on the history of mesalamine use. It would be useful to add information on other treatments including biologic therapy.

As shown in Table 4, remission was more common in the non-EIM group (n = 127, 52.9%) than in the EIM group (n = 8, 25.8%) (p = 0.004). From the study it is difficult to verify, which patients were in remission. It is also not known if lack of adherence or therapy non-complience wast the risk factor for EIM.

·

Basic reporting

The language is, for the most part, fine, but here and there you find are surprising linguistic mistakes.

The literature references are fine, and sufficient background is provided.

The article structure, figures, and tables are fine.

The manuscript is self-contained.

Experimental design

The study represents primary research within the Aims and Scope of the journal.

The research question is fairly well defined, however, it consists of a combination of questions, including 1) to report the prevalence and type of EIMs in China; 2) examining the degree of association between EIM and clinical outcomes; and 3) determining risk factors associated with EIMs. These questions have been addressed in the literature previously, but not well enough for the Chinese population.

There are some caveats with regards to the methodological work (see comments below).

Methods description lacks enough detail in some parts.

Validity of the findings

The study as such and reporting the data it has generated are both justified.

Data is statistically sound, but I would not say robust.

Conclusions need to be adjusted.

Additional comments

This is a well written manuscript examining the prevalence of extraintestinal manifestations (EIMs) of ulcerative colitis (UC) in 271 patients in Shanghai, followed for a median of 13 years. In addition, EIM-associated features of UC are presented, as well as clinical outcomes for UC in patients with, as compared to without, EIMs. Inflammatory bowel disease (IBD), including ulcerative colitis, is on the rise in China, and it is interesting to report on a number of aspects of IBD in this population, including EIM, albeit that numerous similar reports have been published previously from other parts of the world.

The statistical work seems to be correct, but there are other methodological caveats that are problematic (see below).

Major comments:

1. The authors discuss EIM-associated features as if there is a direct causal relationship between them. For example, their data show that anemia is associated with EIMs, and they conclude that anemia should be treated in order to prevent the development of EIM and improve disease prognosis. A more likely relationship is that a high level of colonic inflammation increases the risk for EIMs as well as for anemia, but treating the anemia as such will probably have little impact on the EIMs. One could be very speculative and suggest that relative systemic hypoxia could increase the risk for EIMs, but this is quite far-fetched. Thus, my suggestion is that the authors modify the text so that the various features that are shown to be more common in patients with EIMs or the other way around are described as "associated", but not "contributing" or that there is an "interaction [between anemia and EIM] responsible for the development of EIM".

2. The same goes for the relationship between EIMs and clinical outcomes, i.e. write "associated with" rather than "EIMs having an impact on clinical outcomes" or "EIM can increase the risk of poor clinical outcomes, such as lower remission rates..".

3. The criteria or definitions of EIMs are not well described. Please describe in greater detail how the different EIMs were defined and diagnosed.

4. The patients were followed for a median of 13 years. Table 1 presents the frequency for a number of features including number of EIMs, history of surgery, weight loss, steroid-treatment, treatment with "immunomodulators", etc. The Table-heading says that these are "Baseline characteristics" but nevertheless the table includes data for history of surgery. It is unclear what timepoint these data refer to: First presentation, at the time-point when the medical records were reviewed, or end of follow-up? Are the numbers cumulative or cross-sectional? Are the data for EIMs before, during or after the stated treatment? Do the treatment-numbers refer to "current" treatment or "ever treated with". For the uni- and multivariate association analyses (Table-title reads "univariable" analysis), were the tested factors aligned in time with the EIMs or are they registered as "ever"? Please clarify.

Minor comments:

The Discussion is too wordy, may be shortened.

Row (R) 79-80: How was "poor compliance" defined and why were these patients excluded? What does "underlying disease" mean in this context, and why were these patients excluded? What does "impaired general health" mean in this context, and why were these patients excluded?

R89: Hb and Alb were collected. Why not CRP?

R90: Define what secondary and primary EIMs are.

R93: How was "remission" defined? If "refractory clinical symptoms" includes more than 5 stools per day, it gives the impression that 5 stools/day or less is considered as remission. Is this the case?

R94: In Table 1 the % of patients on "immunomodulators" was 12.5%. In the text the authors discuss "immunosuppressive agents" and "biologics". Clarify what is meant by these terms. How many patients were/had been treated with 5-ASA and biologics, respectively?

R96: The authors define moderate anemia as <90 g/L. Present the cut-offs you used for mild and severe anemia too. Same cut-offs for males and females?

R118: Pancolitis is a subtype of extensive colitis which is the correct term for E3.

R122: How was "rash" defined? A rash can be non-related to UC; caused by medication, or potentially related to UC. Did not any patients present with erythema nodosum or pyoderma gangrenosum?

R123: Arthritis should be divided into Type I and II if possible.

R125: 1/271 is 0.4% (correct in Table 2 but not the text).

R140: "Cure" as a clinical outcome of UC I would suggest to avoid.

R141: "Bleeding" is one of the most common manifestations of UC, thus, defining it as a "serious complication" is not adequate. Crohn's disease should not be defined as a complication of UC (Table 3). "Iris" should be "iritis" and included in the text in addition to the table.

R181: Swiss -> Switzerland

Table 5 and 6: Explain how data were collected - were these factors present at the time of an EIM, or are they not connected in time?

Reviewer 4 ·

Basic reporting

Clear and easy understood.

Experimental design

Retrospective cohot study.

Validity of the findings

no comment

Additional comments

1. The title may have been over-stated, because 1) UC patients from two large general hospitals in Shanghai only, 2) this is a two-center study, and 3) the sample size is relatively small. It would be difficult to generalize the findings to UC patients of eastern China.
2. This is a follow-up study on secondary EIM occurrence in UC patients. So please specify secondary EIM in the title. Prevalence is not appropriate here, it should be cumulative incidence rate. The authors may need to consult an epidemiologist.
3. It would be helpful to specify how UC and EIM are diagnosed, because details are unclear in the paper.
4. In follow up studies, time to event (secondary EIM) is also import. I am confused that why the authors did not provide data on this variable. And survival and cox regression analyses are appropriate in this paper. Statistical methods used are not detailed enough, for example, line 102-104, t-test was used to compare the difference between ?. Methods for selecting significant predictors in multiple analyses are not described. I feel it is unnecessary to mention the statistical methods used in this paper are similar to one previous study.
5. The sample size is too small, particularly those with secondary EIM. Please consider its impact on multiple analysis.

---

## Round 0.2 · Minor Revisions

The paper still has some issues that need to be addressed. Please revise it again following the comments of the reviewers.

Reviewer 1 ·

Basic reporting

Thank you for the thorough and clear revision of the manuscript. The introduction is relevant to the study and the hypothesis is clearly stated. The results are also relevant to the hypothesis

However, I have two comments on the introduction:
-is it possible to describe the Primary and secondary EIM in the introduction?
- in the hypothesis can you state that you are investigating the secondary EIM for Ulcerative colitis?
Additionally, the incidence here is not a cumulative incidence. It can be just that incidence of secondary EIM in Ulcerative colitis patients ....and complete the hypothesis.

Experimental design

The aim is clearly stated thank you and the research question is well defined . Methods are clearly stated and well-detailed. Thank you for providing definitions for each criteria.

line 80 :Just a comment on the exclusion criteria: why do you exclude FAP if you have decided from the beginning to study UC patients. You can exclude UC patients who later had an indeterminate colitis diagnosis. you can also say that you have excluded primary EIM.

In line 116 you can state it as a first,second or third degree relative in stead of immediate relative (according to what you have studied)

Again, thank you for a very clear writing for the methods used and adding the statistical test clearly.

Validity of the findings

Results section is clearly-written, fine-tuned based on statistically sound and controlled methods. The flow chart is

However, i have some comments:
lines 150-160: describes the studied EIM population. I suggest that you write all the data here and remove the table 2 to avoid repetition.

In lines 164-171: remove test from 164-166(it is more a methodology sentence.
keep text from 166 ,starting (in the entire cohort). then divide the complications according to EIM group or not and do the same in the table 3
then describe only the serious complications or the most common complications for the EIM only in the text. The study is eventually about EIM.
please don't forget to add the gastrointestinal bleeding in table 3 after modifying it.

Table 4 is very clear thank you. can you please add to the footnote what were the serious complications.

In results line 187: the age at diagnosis (can you add ≥ 50) and standardize the ≥ all over the text such as in line 210 and 278

table 5 and 6 please keep them. However, i wonder why you tested for relapse in table 6 while it was not significant as a risk factor in table 5

line 207 this is an incidence not a cumulative incidence as far as i know.Please standardize this all over the text starting from the abstract to conclusion.

line 235 don't state results again in the discussion.
line 251 how does it relate to the bowel disorder

line 272, this is not a strength of the study. You should write it in a better form.

Additional comments

Thank you for this nicely written manuscript. You were very through in following the comments and the text have evolved markedly.
Again thank you

Reviewer 2 ·

Basic reporting

I would suggest one more English language spellcheck

Experimental design

No comment

Validity of the findings

It is a pitty that part of the databases were updated, which in consequence led to loss of some data. However it is worth presenting the results of the study to readers

Additional comments

I would recommend changing the term biologics to probiotics as it seems that none of the patients at that time were receiving immunobiologics such as anti-TNF monoclonal antibodies. Authors should perhaps mention this to clarify the terms and not to mislead the readers

·

Basic reporting

OK

Experimental design

OK

Validity of the findings

OK

Additional comments

Congratulations on addressing the comments thoroughly. Two remaining comments:

Reviewer 3:

Major 4: The title of the table cannot be "Baseline" if it refers to the most recent follow-up. Rename to "Patient characteristics at the most recent follow-up visit".

Minor 6: Probiotics should not be defined as Biologics although in a strict meaning they are. Name the "Probiotics" and use the term "Biologics" only for monoclonal antibody drugs.

---

## Round 0.3 · Minor Revisions

Thank you for your detailed revision. The paper still has some minor issues. Please revise it accordingly.

Reviewer 1 ·

Basic reporting

Thank you for editing your text. The hypothesis is well stated as well as the aim of the study. Raw data is shared and data is presented in tables/figure.
However, There is still some repetition of results both in tables and in text. I still suggest dropping table 2 or changing its format similar to table 3 (compare EIM and non EIM)

some English terms are inappropriately used : e.g. relationships (line 108) and contributing (lines 184 ,193) could be association
now (line 99) when writing the methods should define the date of the end of the study

May be provide a reference to the Montreal classification system( line 92)

Experimental design

research question is well defined and sufficient details are provided in the methods section
- I wonder what respectively in line 132 refers to?
- loss of weight : I find it difficult to add loss of 5 kg during the course of UC a significant factor (which is according to your calculation can be 7 years)

Validity of the findings

The study aimed at evaluating the incidence of EIM in china but it didn't see this result reported in the results section, instead it gets reported in the discussion (line)

- line 145: the median age at diagnosis (is it for males or for all population)please make it clear.

- line 146: I wanted to see the demographics of the EIM patients: how many were males, their ages ,the disease duration (at the end this is the target population that the text addresses)

- I still find table 2 as a repetition of the text (lines 152-155). can you minimize the data here and refer to the table.

-Table 3 is very nice please keep it

- can you move line 167 to methods (the definition of serious complications)

-lines 168: add a little about the EIM patients complications ( this is your target population)

-lines 172-173: sounds like methods not results.

-table 4 : serious complications EIM (10 patients) ? where did that come from.When you count it from table 3 and read it in the text,they are only 4 patients!!!

-Table 4 : please mark which comparison used the chi square and which used the fisher's exact.

- Table 6: you mention in the text that you tested in table 5 for possible risk factors (univariate) then proceeded only with those significant into the multivariate analysis. Why did you test for relapse in the multi-variate analysis while it was not statistically significant in the univariate analysis

-the discussion contains a lot of repetition of results, this is not appropriate.

- the strengths of the study should be revised.

Additional comments

Thank you for the opportunity to re-revise your text. The text is getting better but it needs more careful revision in the light of the reviewers comments. I understand the difficulties with the English language but some terms are inappropriate.

Reviewer 2 ·

Basic reporting

This study is now adequatly presented

Experimental design

No comment

Validity of the findings

No comment

Additional comments

Thank you for all amendments and revisions

---

## Round 0.4 · accepted · Accept

I am satisfied with your revision. The paper can be accepted now.